# α-Synuclein Induces Neuroinflammation Injury through the *IL6ST-AS*/STAT3/HIF-1α Axis

**DOI:** 10.3390/ijms24021436

**Published:** 2023-01-11

**Authors:** Danyu Lin, Han Zhang, Jieli Zhang, Kaixun Huang, Ying Chen, Xiuna Jing, Enxiang Tao

**Affiliations:** 1Department of Neurology, The Sun Yat-Sen Memorial Hospital, Sun Yat-Sen University, Guangzhou 510080, China; 2Department of Neurology, The Eighth Affiliated Hospital, Sun Yat-Sen University, Shenzhen 518033, China; 3Guangdong Provincial Key Laboratory of Malignant Tumor Epigenetics and Gene Regulation, Sun Yat-Sen Memorial Hospital, Sun Yat-Sen University, Guangzhou 510120, China; 4Guangdong Province Key Laboratory of Brain Function and Disease, Zhongshan School of Medicine, Sun Yat-Sen University, Guangzhou 510120, China

**Keywords:** α-synuclein, Parkinson’s disease, *IL6ST-AS*/STAT3/HIF-1α axis, neuroinflammation

## Abstract

The aggregation of α-synuclein (α-syn) promotes neuroinflammation and neuronal apoptosis, which eventually contribute to the pathogenesis of Parkinson’s disease (PD). Our microarray analysis and experimental data indicated a significant expression difference of the long noncoding RNA *IL6ST-AS* and its anti-sense strand, *IL6ST*, in α-synuclein-induced microglia, compared with unstimulated microglia. IL6ST is a key component of the IL6R/IL6ST complex in the microglial membrane, which recognizes extracellular inflammatory factors, such as IL6. Studies have shown that the binding of IL6 to the IL6R/IL6ST complex could activate the JAK2-STAT3 pathway and promote an excessive immune response in glia cells. Meanwhile, the phosphorylation and activation of STAT3 could increase the transcription of *HIF1A*, encoding a hypoxia-inducible factor related to cytotoxic damage. Our results indicated that the overexpression of IL6ST-AS induced by exogenous α-synuclein could inhibit the expression of IL6ST and the activation of JAK2-STAT3 pathway in HMC3 cells. In addition, a reduction in STAT3 resulted in the transcription inhibition of *HIF1A* and the acceleration of oxidative stress injury in SH-SY5Y cells co-cultured with α-synuclein-induced HMC3 cells. Our findings indicate that *IL6ST-AS* is an important factor that regulates microglia activation and neuronal necrosis in the progression of PD. In the HMC3 and SH-SY5Y cell co-culture system, the overexpression of *IL6ST-AS* led to microglial dysfunction and neurotoxicology through the *IL6ST-AS*/STAT3/HIF-1α axis. Our research revealed the relationships among α-synuclein, IL6ST, STAT3, and HIF-1α in the pathological process of PD and provided a new inflammation hypothesis for the pathogenesis of PD.

## 1. Introduction

The pathological α-synuclein (α-syn) oligomer or polymer can be released from infected dopaminergic neurons and passed into healthy neurons and microglia [1]. Recent research indicates that microglia could play an important regulatory role in the pathogenesis of Parkinson’s disease (PD). The activation of microglia can promote the anti-inflammatory effect and the degradation of pathological protein precipitates. On the other hand, it also accelerates the pro-inflammatory processes and contributes to the inflammation-associated cytotoxic damage [2,3]. In our previous studies, we used α-synuclein transgenic SH-SY5Y cell-derived exosomes to stimulate the microglia, and the results revealed the autophagy–lysosome pathway dysfunction and the reduced degradation of pathological α-synuclein in microglia [4,5].

Microglia are important immune cells in the central nervous system (CNS). They play an important role in neuroinflammation, neuron repair, and regeneration. A considerable number of studies suggest that microglia participate in the progression of PD [6]. On the one hand, microglia can promote anti-inflammatory effects, leading to the clearance of pathological unfolded proteins through the autophagy-lysozyme pathway [3]. On the other hand, microglia participate in the pro-inflammatory response and release a series of inflammatory cytokines, complement, chemokines, and free radicals, which increase the severity of the pathology and neuron injury [6,7]. However, the neuroinflammatory mechanism elicited by the presence of extracellular α-syn is unclear.

Based on our microarray analysis, we found significant upregulation of long noncoding RNA (lncRNA) Interleukin 6 cytokine family signal transducer—Antisense (IL6ST-AS) in the cerebrospinal fluid of patients with PD. Moreover, we observed *IL6ST-AS* overexpression in human microglial clone 3 (HMC3) cells treated with extracellular α-synuclein. Gene database prediction suggested that *IL6ST-AS* might be related to the transcription of its anti-sense strand, *IL6ST* (encoding interleukin 6 cytokine family signal transducer). IL6ST is a key component of the interleukin 6 receptor (IL6R)/IL6ST complex in the microglial membrane. This complex is responsible for inflammatory signal reception and the related molecular pathway activation [8,9]. Previous studies demonstrated that the IL6/IL6R/IL6ST complex could attenuate the activation of the Janus kinase 2 (JAK2)-signal transducer and activator of transcription 3 (STAT3). The combination of IL6 and IL6R also had an impact in activating nuclear factor kappa B (NF-κB) pathway via regulating JAK2-STAT3 pathway [10,11,12].

The present study aimed to find the key molecular targets in the process of microglia–neuron interaction and to study the relationships among α-synuclein, *IL6ST-AS*, IL6ST, STAT3, and hypoxia-inducible factor alpha (HIF-1α) in the pathological process of PD. The results showed that the JAK2-STAT3 pathway and its downstream transcript regulatory target, HIF-1α, were consistently inhibited in the pathological α-synuclein-induced HMC3 cells and SH-SY5Y cells (neuroblastoma cells) co-culture system. Moreover, HIF-1α-related oxidative stress injury was more severe in co-cultured SH-SY5Y cells than in single-cultured SH-SY5Y cells, even though they were both exposed to α-synuclein. We observed the up-regulation of IL6 expression outside α-synuclein-induced HMC3 cells (Appendix A) and the inhibition of STAT3-HIF-1α axis in α-synuclein-induced HMC3 cells. Considering the relationship between the IL6/IL6R/IL6ST complex and the JAK2/STAT3 pathway, we suggested that the gene regulatory network of IL6ST-AS and IL6ST might be involved in the inhibition of the STAT3-HIF-1α axis in HMC3 cells induced by α-synuclein.

## 2. Results

### 2.1. α-Synuclein Decreased the Expression of IL6ST by Regulating IL6ST-AS in HMC3 Cells

The microarray analysis results (Figure 1A–D) identified a significantly overexpressed lncRNA, *IL6ST-AS*, in the cerebrospinal fluid of patients with PD (Figure 1E). Considering the potential connection between *IL6ST-AS* and its downstream gene, *IL6ST*, we screened the microarray data and found that *IL6ST* was stably downregulated in the cerebrospinal fluid of patients with PD (Figure 1F). To explore the detailed mechanism of *IL6ST-AS* and *IL6ST*, we treated the HMC3 cells with α-synuclein to build a PD-related inflammation model. Compared with the control group, we detected a marked increase in the *IL6ST-AS* in the α-synuclein-treated group (Figure 1G). Interestingly, although the *IL6ST* mRNA level was upregulated in the α-synuclein-induced HMC3 cells (Figure 1H), the protein level of IL6ST was decreased in the α-synuclein-induced group (Figure 1I,J). In situ fluorescence showed that the majority of *IL6ST-AS* was located in the cytoplasm of the HMC3 cells (Figure 1K). According to this finding, we hypothesized that α-synuclein upregulated *IL6ST-AS*, which might upregulate the transcript level of *IL6ST* but decrease its translation in the cytoplasm.

We concluded from Figure 2A, B that the expression of IL6ST was decreased in α-synuclein-induced HMC3 cells and *IL6ST-AS* UP HMC3 cells. Considering that there might be a cis-acting regulatory relationship between *IL6ST-AS* and *IL6ST*, we used competing endogenous RNA (ceRNA) network data to predict an intermediate regulatory microRNA. ceRNA analysis predicted that *IL6ST-AS* could affect the transcription of *IL6ST* by modulating the expression of mir-488-3p (Figure 2C). A subsequent fluorescein test showed that mir-488-3p could combine with both *IL6ST-AS* and *IL6ST* at the same 3′ UTR binding site (Figure 2D–G). This analysis predicted that *IL6ST-AS* would increase the mRNA transcript level of *IL6ST* by inhibiting the binding of *IL6ST* and mir-488-3p, thereby relieving the mir-488-3p-induced blockade of IL6ST translation.

### 2.2. Overexpression of IL6ST-AS Resulted in Inhibition of JAK2-STAT3 and NF-κB Pathways

Next, we investigated the effect of the overexpression of *IL6ST-AS*. By regulating the expression of *IL6ST*, *IL6ST-AS* would have an impact upon the JAK2-STAT3 and NF-κB pathways, which are the downstream signal pathways of the IL6/IL6R/IL6ST complex. In HMC3 cells treated with α-synuclein or overexpressing *IL6ST-AS* after plasmid transfection, activation of the JAK2-STAT3 and NF-κB pathway was inhibited compared with the control group. In *IL6ST-AS* knockdown (KD) HMC3 cells stimulated by α-synuclein, we detected increased levels of phosphorylated (p)-JAK2, p-STAT3, and NF-κB compared with those in the α-synuclein-induced group (Figure 3A–D,G,H). This result illustrated that *IL6ST-AS* is a vital factor modulating the activation of both the JAK2-STAT3 and NF-κB pathways in HMC3 cells.

### 2.3. Overexpression of IL6ST-AS in HMC3 Cells Affected the STAT3-HIF-1α Axis Inhibition in HMC3 Cells and SH-SY5Y Cells

Next, we focused on the downstream target of both the JAK2-STAT3 and NF-κB pathways in *IL6ST-AS*-overexpressing HMC3 cells. Inhibition of the JAK2-STAT3 pathway led to decreased HIF-1α levels in the IL6ST-AS overexpression group and the α-synuclein-induced group. The expression of HIF-1α increased slightly in the *IL6ST-AS* KD HMC3 cells stimulated by α-synuclein. This implies that inhibition of HIF-1α could be reversed by downregulation of *IL6ST-AS*, even in α-synuclein-induced HMC3 cells (Figure 3E,F).

When cultured with *IL6ST-AS*-overexpressing HMC3 cells or α-synuclein-induced HMC3 cells (Figure 4A), the levels of p-STAT3 and HIF-1α in SH-SY5Y cells decreased compared with those in the control SH-SY5Y cells. This change tendency of p-STAT3 and HIF-1α was different from that in α-synuclein-induced SH-SY5Y cells (Figure 4B–G). This indicates that the *IL6ST-AS*-dependent JAK2-STAT3 pathway inhibition in HMC3 cells might be an important regulatory process modulating the expression of HIF-1α in co-cultured SH-SY5Y cells. In particular, the decrease in HIF-1α was reversed in SH-SY5Y cells co-cultured with *IL6ST-AS* KO HMC3 cells, regardless of whether they were stimulated with α-synuclein or not.

### 2.4. α-Synuclein Inhibits Oxidative Stress Repair in the HMC3-SH-SY5Y Cell Co-Culture System by Modulating the IL6ST-AS/STAT3/HIF-1α Axis

We detected the ROS content to measure the level of oxidative stress in the in vitro α-synuclein-induced model. After stimulation by exogenous α-synuclein, the cellular ROS content increased in the first few hours and then tended to decrease later (Appendix A). Approximately 24 h after stimulation, the ROS content was scavenged by 27% in the α-synuclein-induced SH-SY5Y cells. These results illustrated that continuous α-synuclein induction could rescue the oxidative stress injury. This phenomenon was also observed in the SH-SY5Y–HMC3 cell co-culture system. The ROS content was decreased in SH-SY5Y cells co-cultured with α-synuclein-induced HMC3 cells compared with that in SH-SY5Y cells co-cultured with non-induced HMC3 cells. However, the decrease in ROS content induced by α-synuclein in co-cultured SH-SY5Y cells was markedly less than that in SH-SY5Y cells cultured alone without α-synuclein treatment. This suggests that co-culturing with HMC3 cells would have a negative impact on the oxidative stress repair promoted by α-synuclein (Figure 5A,B).

In the co-culture system, upregulating the expression of the *IL6ST-AS* in HMC3 cells inhibited the oxidative-stress-protection effect in co-cultured SH-SY5Y cells, regardless of whether the HMC3 cells were stimulated by α-synuclein. By contrast, in the co-culture system, decreasing the expression of IL6ST-AS in HMC3 cells dramatically promoted the oxidative stress repair dramatically in co-cultured SH-SY5Y cells, particularly, when SH-SY5Y cells were co-cultured with α-synuclein-induced HMC3 cells (Figure 5C–F).

## 3. Discussion

In this study, we stimulated HMC3 cells with α-synuclein and revealed the inhibition of IL6ST-AS/STAT3/HIF-1α Axis in HMC3 cells. When SH-SY5Y cells were co-cultured with α-synuclein-treated HMC3 cells, the expression of HIF-1α and the activity of scavenging ROS were reversed in SH-SY5Y cells.

α-synuclein-induced models showed that pathological α-synuclein could be released from infected dopaminergic neurons to induce microglia-related inflammation injury [7]. Our previous research found that exosomes derived from *SNCA* (encoding α-synuclein) transgenic neurons delivered pathological α-synuclein and transcriptional information to microglia. The *SNCA* transgenic neuron-delivered exosomes suppressed autophagy via the miR-19a-3p/phosphatase and the tensin homolog (PTEN)/mechanistic target of the rapamycin kinase (mTOR) axis in recipient microglia [6]. Accordingly, we speculated there might be a connection between the immune response of abnormal microglia and the neuronal damage in the pathogenesis of PD [13]. This present study aimed to find the key molecular targets in the process of microglia–neuron interaction.

We identified a significantly differently expressed lncRNA, *IL6ST-AS*, in the cerebrospinal fluid from patients with PD using microarray analysis. *IL6ST-AS* expression in α-synuclein-induced HMC3 cells was also increased. However, the transcript levels of its downstream regulatory gene, *IL6ST*, showed the opposite tendency in the cellular model and in the cerebrospinal fluid from patients with PD. Combined with the luciferase assay results, we hypothesized that *IL6ST-AS* might be a cis-acting regulatory factor of *IL6ST*. Our results indicated that the *IL6ST-AS*/mir-488-3p/*IL6ST* gene-regulatory network affected the translation of IL6ST. However, in α-synuclein-induced HMC3 cells, the *IL6ST* transcript level was increased, but the protein level was decreased. The transcript level of the upstream gene, *IL6ST-AS*, was inhibited, which could not result from a lower protein translation or the rapid degradation of protein after translation. However, the expression of IL6ST did not decrease significantly in the control group. Thus, there might be some regulatory elements in the primary transcript of *IL6ST* that led to decreased protein translation.

This decrease in IL6ST in α-synuclein-induced HMC3 cells was consistent with that in the cerebrospinal fluid of patients with PD; however, the mechanism of this tendency, as mentioned above, was not the same. The degradation of IL6ST’s transcription in cerebrospinal fluid karyocyte was taken into account [14]. Our results demonstrated that *IL6ST-AS* is an upstream regulatory factor of *IL6ST*, and it indeed decreased the expression of IL6ST in HMC3 cells.

The upregulation of *IL6ST-AS* not only influenced the expression of IL6ST but also affected the downstream molecular signaling pathway associated with IL6ST. Increased IL6 levels were confirmed in PD inflammatory models, such as lipopolysaccharide (LPS)-induced microglia or rotenone-induced microglia [15,16]. In our study, the endogenous IL6 was stably increased in α-synuclein-treated HMC3 cells (Appendix A). However, as the target pathway of the IL6/IL6R/IL6ST complex, activation of the JAK2-STAT3 pathway was inhibited in α-synuclein-treated HMC3 cells. The experimental results suggested that *IL6ST-AS*-related IL6ST inhibition might be an important regulatory factor that accounts for the dephosphorylation of the JAK2-STAT3 pathway. The accumulation of IL6 and IL1β is an inflammatory response process in microglia against the inflammatory cells and inflammatory factors in the micro-environment [17]. Depending on the combination of IL6 and the IL6R/IL6ST complex, downstream pathways, such as the JAK2-STAT3 and NF-kB pathways, would be activated. According to the results shown in Figure 2 and Figure 3, the inhibition of the JAK2-STAT3 pathway in HMC3 cells was accounted for by the IL6ST-AS-related IL6ST inhibition induced by α-synuclein, an extracellular neurotoxic factor. This indicated that the overexpression of *IL6ST-AS* in HMC3 cells had a negative impact on the α-synuclein-related anti-inflammatory properties.

The phosphorylation and activation of STAT3 influences the transcription of *HIF1A*, which affects hypoxia-mediated cell damage [18,19]. When the inflammatory response was intensified, the JAK2-STAT3 pathway was activated in HMC3 cells. Phosphorylated STAT3 would then be transferred into the nucleus, where it regulates the transcription regulatory of *HIF1A* [19]. A study reported that the colocalization of phosphorylated STAT3 and HIF-1α could be observed within the nucleus under the conditions of inflammation activation [19]. Our results illustrated that α-synuclein could decrease the expression of HIF-1α in HMC3 cells and their co-cultured SY-SH5Y cells. This observation was closely related to α-synuclein-induced IL6ST-AS overexpression and IL6ST/JAK2/STAT3 signal pathway inhibition.

Research has shown that HIF-1α modulates the oxidative stress response during the progress of neurodegenerative diseases [20]. The overexpression of HIF-1α in neurons would result in neurotoxic damage, including the dysregulation of neuronal growth, proliferation, and repair [21]. An epidemiological study suggested that HIF-1α had a certain correlation with the onset of PD [22]. Subsequent molecular research reported that HIF1α might participate in the occurrence and development of PD through transcriptional regulation of PD disease-related genes, such as LRRK2 (encoding leucine rich repeat kinase 2) [23,24].

In addition to the negative impact of HIF-1α overexpression, some studies indicated HIF-1α could be a neural protective factor in the progression of PD and Alzheimer’s disease (AD) [25,26,27]. In vitro models of AD and PD verified that stimulation by certain neuroprotective drugs, such as multifunctional brain-penetrating iron chelator M30 or deferoxamine (DFO), could increase the expression level of HIF-1α to promote the activation of antioxidant pathways and exert neuroprotective effects. In a cellular model of AD, it was observed that an increase in HIF-1α was associated with the degradation of the Aβ protein in hippocampal neurons; however, the detailed mechanism underlying this phenomenon is unclear [28].

In this study, we identified HIF-1α as a protective factor in a HMC3 and SY-SH5Y cell co-culture system. The inhibition of the HIF-1α induced by α-synuclein in the co-culture system was stable; nevertheless, this inhibition was not observed in SY-SH5Y cells cultured alone. The protective effect of HIF-1α was neatly illustrated by the change in the ROS content. We found that the ROS content’s change tendency was stably reversed at 24 h after α-synuclein stimulation in SY-SH5Y cells alone and in the co-culture system. Moreover, the reduction in the ROS content in SY-SH5Y cells was greater than that in the HMC3 cells/SY-SH5Y cell co-culture system. This result implies that α-synuclein stimulation contributed to the repair of cell dysfunction after oxidative stress by inhibiting HIF-1α expression in the co-culture system. Considering the ROS content change in the *IL6ST-AS* overexpression group and the IL6ST-AS KD group, these results demonstrated that α-synuclein inhibited HIF-1α expression and promoted the repair of cell dysfunction after oxidative stress by regulating the expression of *IL6ST-AS*.

In conclusion, α-synuclein can regulate the expression of *IL6ST-AS* and *IL6ST* in HMC3 cells, causing the inactivation of the JAK2/STAT3 signal pathway, thus mediating HIF-1 expression in co-cultured neurons and the accumulation of toxic damage in SY-SH5Y cells. Thus, the inhibition of the *IL6ST-AS*/STAT3/HIF-1α axis in microglia resulting from α-synuclein stimulation led to reduced HIF-1α expression and increased oxidative stress injury in SH-SY5Y cells. There may be a vicious cycle between “neuroinflammation in microglia” and “dopaminergic neuron oxidative stress injury”, which jointly promote neurotoxic damage in the progression of PD. In this vicious cycle, the expression of *IL6ST-AS* in HMC3 cells is an important regulatory link. By regulating the expression of *IL6ST-AS* in HMC3 cells, the inhibition of HIF-1α expression in SH-SY5Y could be reversed, which means the neuroprotective effect of HIF-1α on cellular oxidative stress could be achieved via this vicious cycle.

## 4. Materials and Methods

### 4.1. Patients’ Clinical Data and CSF Specimen Collection

The PD patients were selected from Sun YAT-SEN memorial hospital, Sun YAT-SEN University. They were included into the clinical follow-up from 2015, and the patients’ CSF specimens were collected in 2018 or 2019. We have included the demographic description of the patients in the Appendix A.

The normal controls were collected from the clinical discarded specimen of patients in Sun YAT-SEN memorial hospital (Appendix A). We performed a lumbar paracentesis upon the clinical patients to clarify a diagnosis and collected the CSF for routine clinical biochemical test. Their CSF would be stored in the clinical laboratory for several days. We selected the patients without organic disease in the central nervous system and retrieved their CSF from clinical patients with the patients’ permission.

All these CSF specimens mentioned above were stored in a liquid nitrogen container and transported in a dry ice box.

### 4.2. Microarray and Array Hybridization

Total RNA was extracted from the CSF specimen using Trizol reagent (Takara Bio Inc., San Jose, CA, USA). RNA quantity and quality were measured by NanoDrop ND-1000. RNA integrity was assessed by standard denaturing agarose gel electrophoresis or Agilent 2100 Bioanalyzer. Arraystar Human LncRNA Microarray V4.0 was designed for the global profiling of human LncRNAs and protein-coding transcripts. Sample labeling and array hybridization were performed according to the Agilent One-Color Microarray-Based Gene Expression Analysis protocol (Agilent Technology) with minor modifications. Briefly, each sample was amplified and transcribed into fluorescent cRNA by Quick Amp Labeling Kit, One-Color (Agilent p/n 5190-0442). The labeled cRNAs were purified by RNeasy Mini Kit (Qiagen, Dusseldorf, Germany). The concentration and specific activity of the labeled cRNAs (pmol Cy3/μg cRNA) were measured by NanoDrop ND-1000. In total, 1 μg of each labeled cRNA was fragmented by adding 5 μL 10 × Blocking Agent and 1 μL of 25 × Fragmentation Buffer, and then the mixture was heated at 60 °C for 30 min; finally, 25 μL 2 × GE Hybridization buffer was added to dilute the labeled cRNA. Furthermore, 50 μL of hybridization solution was dispensed into the gasket slide and assembled to the LncRNA expression microarray slide. The slides were incubated for 17 h at 65 °C in an Agilent Hybridization Oven. The hybridized arrays were washed, fixed and scanned using the Agilent DNA Microarray Scanner (part number G2505C).

Agilent Feature Extraction software (version 11.0.1.1) was used to analyze acquired array images. Quantile normalization and subsequent data processing were performed with using the GeneSpring GX v12.1 software package (Agilent Technologies, Santa Clara, CA, USA). Differentially expressed LncRNAs and mRNAs between the two samples were identified through Fold Change filtering.

### 4.3. Cell Culture

HMC3 cells and SH-SY5Y cells were obtained from Procell Life Science &Technology Company (Wuhan, China). HMC3 cells were cultured in Minimum Essential Medium (MEM) with non-essential amino acids (NEAA), containing 10% fetal bovine serum (FBS), 100 U/mL penicillin, and 100 μg/mL streptomycin in a humidified atmosphere at 37 °C with 5% CO_2_. SH-SY5Y cells were cultured in MEM with F12 containing 15% FBS at 37 °C with 5% CO_2_. The cells were subcultured every 2–3 days when confluency reached 70–80%.

SH-SY5Y cells were differentiated in 6-well culture plates using 10 μM retinoic acid (RA) for 3 days, followed by 80 nM 12-O-tetradecanoylphorbol-13-acetate (TPA) for another 3 days.

The co-culture of HMC3 and SH-SY5Y cells was performed using a Transwell co-culture system, in which HMC3 cells (2 × 10^5^ cells) were cultured in the upper layer and SH-SY5Y cells (2 × 10^5^ cells) were cultured in the lower layer.

### 4.4. Preparation of the α-Syn Oligomer

The stock solution of α-Syn monomer was purchased from Genemei Biotech Co., Ltd. (Guangzhou, China), then diluted in PBS buffer, and incubated in a rotary shaker at 100–150 rpm for 72 h at 37 °C to form oligomers. HMC3 cells and differentiated SH-SY5Y cells were treated with 25 μmol/L α-syn oligomer for 24 h.

### 4.5. Establishment of IL6ST-AS Upregulation and Downregulation in HMC3 Cells

Lentiviral vectors hU6-IL6ST-AS-CBh-gcGFP-IRES-puromycin and Ubi-IL6ST-AS-CBh-gcGFP-IRES-puromycin were constructed by GeneChem Co., Ltd. (Shanghai, China). For lentivirus transfection, HMC3 cells were transfected with the two plasmids described above (multiplicity of transfection (MOI), 10:1), in accordance with the protocol of the manufacturer. The transfection medium was replaced with complete medium after 6 h, and the cells were passaged every 3 days for at least 14 days.

### 4.6. Lentivirus Construction

Lentiviruses expressing the full-length human *IL6ST-AS* cDNA and lentiviruses carrying a short hairpin RNA (shRNA) against *IL6ST-AS* were constructed by Shanghai Genechem Company (Shanghai, China). The RNA interference (RNAi) sequence targeting human *IL6ST-AS* was 5′-#ACAACCAAGTACTGCTGAA-3′. The lentiviruses were amplified and titrated in HEK293T cells, in accordance with the instructions of the manufacturer. Lentiviruses containing empty plasmids (vector) and lentiviruses containing non-specific shRNA (scramble) were used as controls.

### 4.7. Plasmid Construction

GV513 and GV493 plasmids were previously constructed and validated by sequencing.

### 4.8. Cell Culture and Transfection

HEK293T cells were cultured in Dulbecco’s Modified Eagle’s Medium (Corning Inc., Corning, NY, USA) with 10% FBS. HEK293T cells were transfected with the aid of Lipofectamine 3000 (Invitrogen, Waltham, MA, USA), in accordance with the instructions of the manufacturer (Appendix A).

### 4.9. RNA Transfection and Interference

The PsiCHECK-2 vector and mimics were synthesized by Hanbio Biotechnology Co., Ltd. Shanghai, China). *IL6ST* 3′ UTR and LncRNA *IL6ST-AS* wild-type (WT) potential microRNA (miR) binding sites were constructed separately into the PsiCHECK-2 vector, named WT (3UTR) and WT (Lnc), respectively. The mutated (Mut) potential binding sites in *IL6ST* 3’ UTR and LncRNA *IL6ST-AS* Mut miR were constructed into Psicheck-2 vector, named Mut (3UTR) and Mut (Lnc) (mutation pattern: A/T and G/C interconversion), respectively.

### 4.10. RNA Extraction and Quantitative Real-Time Reverse Transcription PCVR (qRT-PCR)

Medium was aspirated, and cells were washed twice with phosphate-buffered saline (PBS) at room temperature. RNA was extracted using a Fast Tissue RNA Purification Kit (EZBioscience, Roseville, MN, USA), in accordance with the instructions of the kit. The concentration and quality of the RNA were detected by IMPLEN N60 Touch (Implen, Munich, Germany). Next, cDNA was prepared using the RNA as a template and a PrimeScript™ RT Reagent Kit (Takara, Shiga, Japan), in accordance with the instructions of the manufacturer, and diluted 1:5 in double-distilled water before quantitative real-time PCR (qPCR) plate preparation. All qPCR reactions were performed in 96-well plates, using SYBR^(R)^ Premix Ex Taq^TM^. *ACTB* (encoding β-actin) was used as the housekeeping gene in qPCR for HMC3 cells. Relative expression was calculated using the 2^−∆∆Ct^ method [29]. The primer sequences used are as follows (Table 1).

### 4.11. Protein Extraction and Western Blotting Analysis

After transfection of recombinant lentiviral vectors, HMC3 and SH-SY5Y cells were seeded at different densities depending on treatment. Medium was aspirated, and cells were washed three times with pre-cooled PBS on ice. Cells were then lysed in radioimmunoprecipitation assay (RIPA) lysis buffer (strong) (Cwbiotech, Beijing, China), supplemented with a protease inhibitor cocktail (Cwbiotech) and a phosphatase inhibitor cocktail (Cwbiotech), both at a dilution of 1:100. The proteins in each sample (20–30 μg of protein per lane) were separated using 10% SDS-PAGE. Then, the proteins were transferred to 0.2 μm polyvinylidene fluoride (PVDF) membranes (Merck Millipore, Billerica, MA, USA). After blocking with 5% Bovine Serum Albumin (BSA, Saibao Biotechnology, Yancheng, China), the membranes were incubated overnight at 4 °C with primary antibodies (1:1000). The next day, the membranes were incubated in secondary antibodies (1:5000) for least 1 h at room temperature and visualized using an Omni-ECL™ Enhanced Pico Light Chemiluminescence Kit (Epizyme Biomedical Technology Co., Ltd., Shanghai, China). Glyceraldehyde-3-phosphate dehydrogenase (GAPDH) and Tubulin were used as the internal standards. The quantitative measurements of immunoreactive protein levels were performed using ImageJ software (NIH, Bethesda, MD, USA)

### 4.12. Fluorescence In Situ Hybridization (FISH)

HMC3 cells (2 × 10^5^ cells per well) were incubated in confocal dishes for 24 h. Cells were washed with PBS for 5 min, fixed with 4% paraformaldehyde at room temperature for 10 min, and then washed for 5 min with cold PBS three times. Then, 1 mL of precooled permeabilization solution (PBS containing 0.5% Triton X-100) was added to each well and incubated at 4 °C for 5 min. Then, the cells were washed for 5 min with PBS three times. According to the instruction manuals of the Ribo^TM^ Fluorescent In Situ Hybridization Kit (RiboBio, Guangzhou, China), 200 μL of the pre-hybridization buffer was added to each well and blocked at 37 °C for 30 min. In the dark, a mixed solution of 5 μL 20μM lncRNA FISH Probe and 18S FISH Probe was added to the 200 μL preheated hybridization solution, respectively. Then, 200 μL of hybridization solution with the probe was added into the cells in the wells of the confocal dishes and hybridized overnight at 37 °C.

The following day (day 2), all processes were manipulated in a light avoidance condition. First, hybridization buffers I, II, and III were preheated to 42 °C. The dishes were washed for 5 min with hybridization buffer I three times and with hybridization buffer II and hybridization buffer III once each. The Petri dishes were subsequently washed for 5 min with PBS three times. Then, 4′,6-diamidino-2-phenylindole (DAPI) staining solution was added for 10 min to stain the cell nuclei. Cells were again washed for 5 min with PBS three times. Finally, an anti-fluorescence quencher was used to seal the dishes, and fluorescence confocal microscopy was carried out using a confocal microscope (Zeiss LSM 710, Oberkochen, Germany).

### 4.13. Detection of Intracellular Reactive Oxygen Species (ROS)

Intracellular ROS generation was assessed using an oxidation-sensitive fluorescent probe (Dichloro-dihydro-fluorescein diacetate (DCFH-DA)) in a Reactive Oxygen Species Assay Kit (Beyotime, Shanghai, China). Probes were diluted in gradients to the appropriate concentrations. Cells were incubated with the suitable concentration of DCFH-DA at 37 °C for 20 min, in accordance with the instructions of the kit. The cells were washed with fresh serum-free medium three times. The fluorescence of DCFH was detected using flow cytometry. The excitation and emission wavelengths were 488 nm and 525 nm, respectively.

### 4.14. Statistical Analysis

Each in vitro experiment was performed in triplicate, and all assays were repeated three times. The mean values of three measurements were calculated and used for statistical evaluation. Data are presented as the mean value ± standard deviation (SD). Comparisons between two groups were performed using Student’s *t*-test, whereas comparisons between more than two groups were performed using one-way analysis of variance (ANOVA) using GraphPad Prism 8 (GraphPad Inc, La Jolla, CA USA). *p* < 0.05 was considered statistically significant.

## Figures and Tables

**Figure 1 ijms-24-01436-f001:**
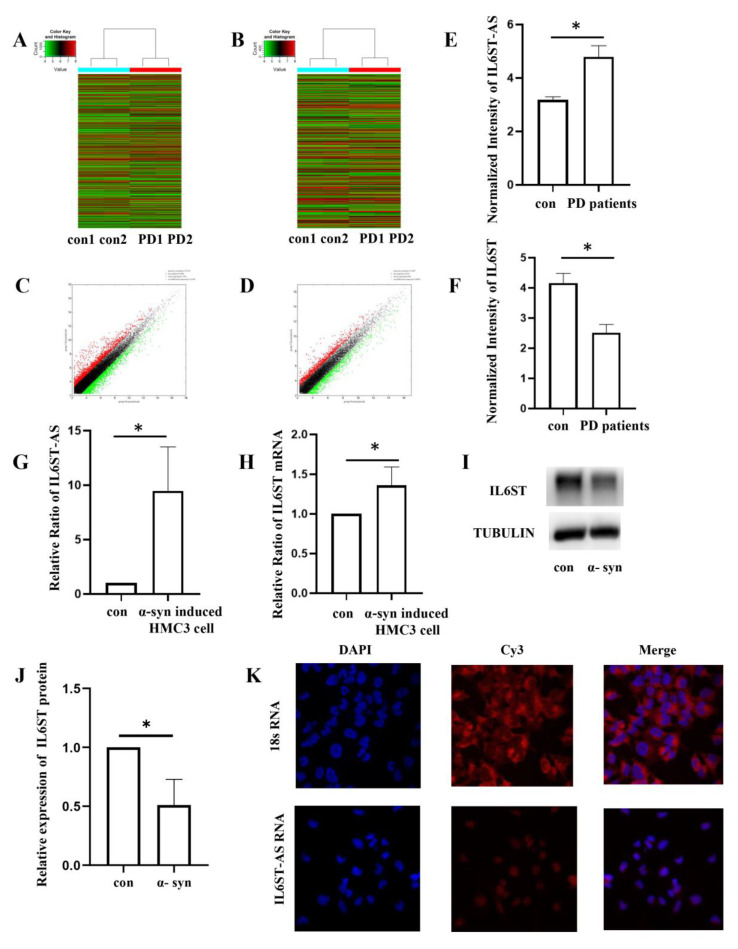
The expression level and sublocalization of IL6ST-AS and IL6ST in PD patients’ specimen and α-syn-induced HMC3 cells. Heat maps of the lncRNA group (**A**) and mRNA group (**B**) showing differentially expressed lncRNAs and mRNAs in the PD patients’ cerebrospinal fluid specimens. The bar code represents the color scale of the log-2-transformed values. Red parts indicate higher expression (FC > 1.5), and green parts indicate lower expression (FC > 1.5). The scatter plots of the lncRNA group (**C**) and the mRNAs group (**D**) were used to assess the variation between the chips. The x-axis and y-axis in the scatter plot represent the normalized signal values of each group (log-2 scaled). The black lines represent the fold change (FC < 1.5). Red points indicate higher expression, whereas green points indicate lower expression. (**E**) Relative expression of IL6ST-AS was detected by qPCR in control group patients and PD patients. (**F**) Relative expression of IL6ST was detected by qPCR in control group patients and PD patients. (**G**) Relative expression of IL6ST-AS was detected by qPCR in control HMC3 cells and α-syn-induced HMC3 cells. (**H**) Relative expression of IL6ST was detected by qPCR in control HMC3 cells and α-syn-induced HMC3 cells. (**I**) The expression of IL6ST was detected by Western blot in control HMC3 cells and α-syn-induced HMC3 cells. (**J**) Quantitative data of IL6ST Western blots. Data = mean ± SD, * indicates *p* < 0.05. (**K**) Sublocalization of IL6ST-AS in HMC3 cells. Taking 18S located in the cytoplasm as the control, most of IL6ST-AS in the HMC3 cells line was located in the cytoplasm.

**Figure 2 ijms-24-01436-f002:**
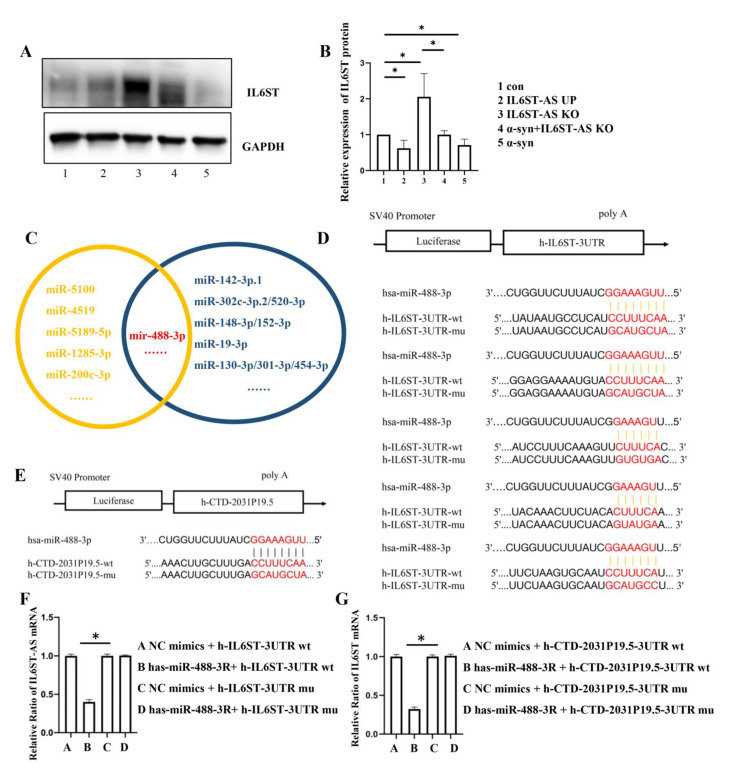
α-synuclein decreased the expression of IL6ST by regulating IL6ST-AS. HMC3 cells were divided into five groups: control (con), IL6ST-AS UP, IL6ST-AS KO, α-synuclein-induced (25 μmol/L) HMC3 cells (α-syn), and α-syn + IL6ST-AS KO. (**A**) The expression of IL6ST was detected by Western blot in HMC3 cells. (**B**) Quantitative data of Western blots in (**A**). Data = mean ± SD, * indicates *p* < 0.05. (**C**) Prediction results for IL6ST-AS lncRNA-binding miRNA and IL6ST mRNA-binding miRNA from two databases, LncBase and TargetScan. (**D**) Predicted miRNA binding sites within human IL6ST. (**E**) Predicted miRNA binding sites within IL6ST-AS (h-CTD-2031P19.5). (**F**) h-IL6ST-3UTR and has-miR-488-3R mimics’ co-transfection obviously restrained luciferase activity by dual-luciferase reporter assay. (**G**) h-CTD-2031P19.5-3UTR and has-miR-488-3R mimics’ co-transfection obviously restrained luciferase activity by dual-luciferase reporter assay.

**Figure 3 ijms-24-01436-f003:**
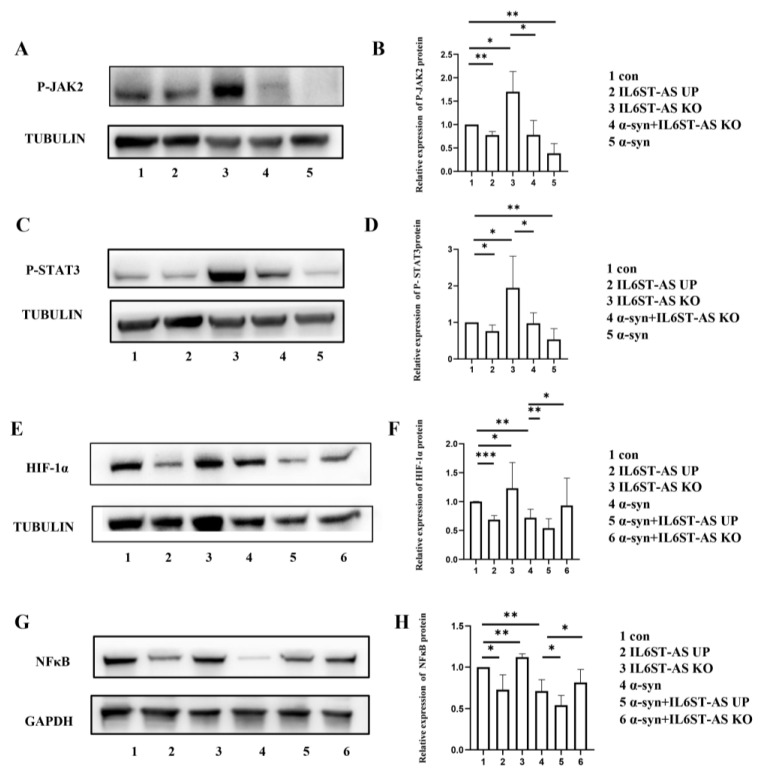
Over-expression of IL6ST-AS resulted in the inhibition of JAK2-STAT3 pathway and NF-κB-related pathway. HMC3 cells were divided into five groups: control (con), IL6ST-AS UP, IL6ST-AS KO, α-synuclein-induced (25 μmol/L) HMC3 cells (α-syn), and α-syn + IL6ST-AS KO. (**A**) The expression of P-JAK2 was detected by Western blot in HMC3 cells. (**B**) Quantitative data of Western blots in (**A**). (**C**) The expression of P-STAT3 was detected by Western blot in HMC3 cells. (**D**) Quantitative data of Western blots in (**C**). HMC3 cells were divided into six groups: control (con), IL6ST-AS UP, IL6ST-AS KO, α-synuclein-induced (25 μmol/L) HMC3 cells (α-syn), α-syn + IL6ST-AS UP, and α-syn + IL6ST-AS KO. (**E**) The expression of HIF-1α was detected by Western blot in HMC3 cells. (**F**) Quantitative data of Western blots in (**E**). (**G**) The expression of NF-κB was detected by Western blot in HMC3 cells. (**H**) Quantitative data of Western blots in (**G**). Data = mean ± SD, * indicates *p* < 0.05, ** indicates *p* < 0.01, *** indicates *p* < 0.001.

**Figure 4 ijms-24-01436-f004:**
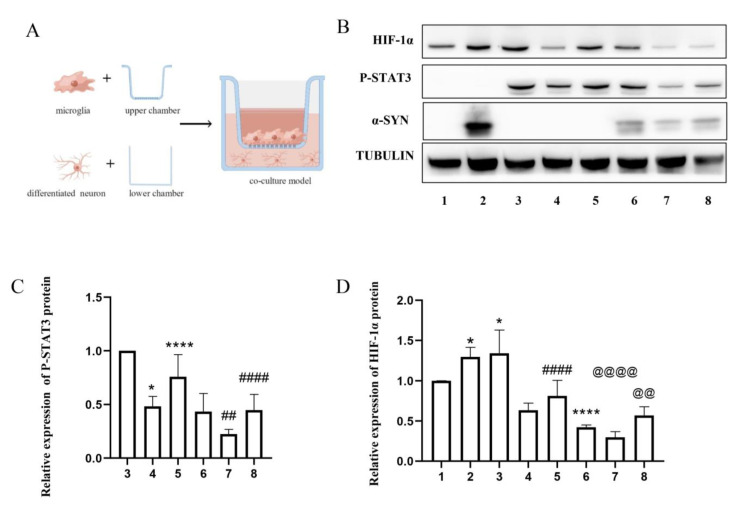
Over-expression of IL6ST-AS in microglia had an effect on the STAT3-HIF-1α axis restraint in neurons. SH-SY5Y cells were divided into eight groups: 1. only SH-SY5H cells, 2. α-synuclein-induced (25 μmol/L) SH-SY5H cells, 3. SH-SY5H cells co-cultured with HMC3 cells, 4. SH-SY5H cells co-cultured with IL6ST-AS UP HMC3 cells, 5. SH-SY5H cells co-cultured with IL6ST-AS KO HMC3 cells, 6. SH-SY5H cells co-cultured with α-synuclein-induced (25 μmol/L) HMC3 cells. 7. SH-SY5H cells co-cultured with α-synuclein-induced (25 μmol/L) IL6ST-AS UP HMC3 cells, and 8. SH-SY5H cells co-cultured with α-synuclein-induced (25 μmol/L) IL6ST-AS KO HMC3 cells. (**A**) Schema of HMC3 cells and SH-SY5H cells co-culture. (**B**) The expression of P-STAT3 and HIF-1α in SH-SY5H cells was detected by Western blot. (**C**) Quantitative data of P-STAT3 Western blots in different groups. (**D**) Quantitative data of HIF-1α Western blots in different groups. Data = mean ± SD, * indicates *p* < 0.05, **** indicates *p* < 0.0001, @@ or ## indicates *p* < 0.01, #### or @@@@ indicates *p* < 0.0001.

**Figure 5 ijms-24-01436-f005:**
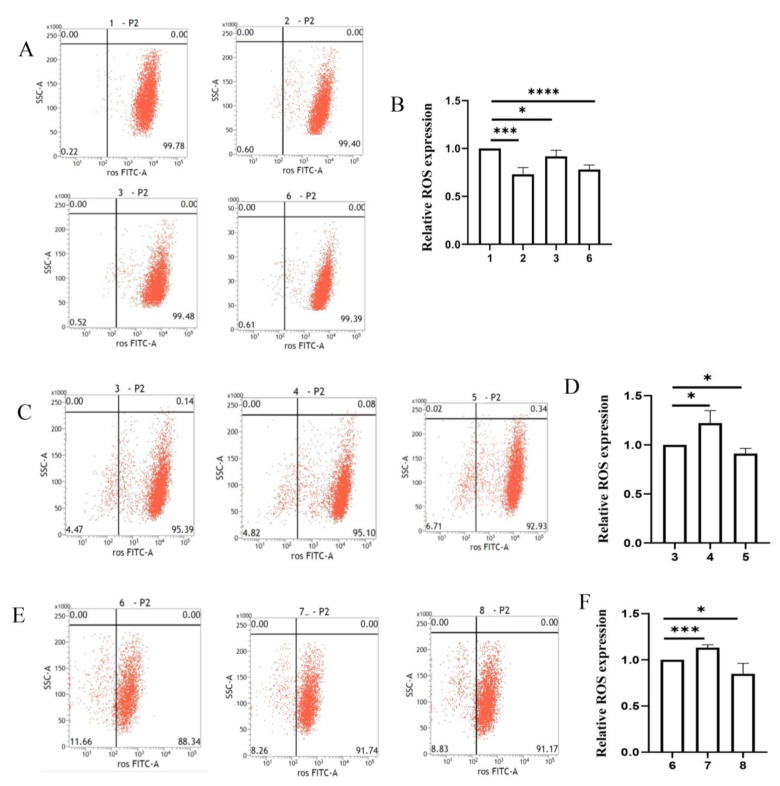
α-synuclein reduced the ROS content in neuron–microglia co-culture system by modulating IL6ST-AS/STAT3/HIF-1α axis. SH-SY5Y cells were divided into eight groups: 1. only SH-SY5H cells, 2. α-synuclein-induced (25 μmol/L) SH-SY5H cells, 3. SH-SY5H cells co-cultured with HMC3 cells, 4. SH-SY5H cells co-cultured with IL6ST-AS UP HMC3 cells, 5. SH-SY5H cells co-cultured with IL6ST-AS KO HMC3 cells, 6. SH-SY5H cells co-cultured with α-synuclein-induced (25 μmol/L) HMC3 cells. 7. SH-SY5H cells co-cultured with α-synuclein-induced (25 μmol/L) IL6ST-AS UP HMC3 cells, and 8. SH-SY5H cells co-cultured with α-synuclein-induced (25 μmol/L) IL6ST-AS KO HMC3 cells. (**A**, **C**, **E**) Flow cytometric analysis of ROS levels in different SH-SY5Y cells. (**B**,**D**,**F**) Quantitative data of ROS fluorescence intensities in different SH-SY5Y cells. Data = mean ± SD, * indicates *p* < 0.05, *** indicates *p* < 0.001, **** indicates *p* < 0.0001.

**Table 1 ijms-24-01436-t001:** The primer sequences of IL6ST-AS and IL6ST.

Primer	Sequences (5′ to 3′)
IL6ST-AS-F	GCCTGAGTGAAACCCAATG
IL6ST-AS-R	CTAGCCAAGTCTGCAACGT
IL6ST-F	GCACCTCCATACTTGGGCTCTT
IL6ST-R	TGTACGGCAAGGCGGCTAC
β-actin-F	CTACCTCATGAAGATCCTCACCGA
β-actin-R	TTCTCCTTAATGTCACGCACGATT

## Data Availability

Data available on request from the authors.

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
