# Peer review of "α-Synuclein Induces Neuroinflammation Injury through the *IL6ST-AS*/STAT3/HIF-1α Axis"

_ijms, 2023, doi:10.3390/ijms24021436_

Round 1
Reviewer 1 Report
Reviewer comments and suggestions
The aggregation of α-synuclein (α-syn) promotes neuroinflammation and neuronal apoptosis, which finally contribute to the pathogenesis of Parkinson's disease (PD). In this study, authors observe the relationships among α-synuclein, IL6ST, STAT3, and HIF-1α in the pathological process of PD and provided a new inflammation hypothesis for the pathogenesis of PD.
Overall, the manuscript was well written. However, a few concerns/comments needed to be explained/modified.
- Line 25-26 Please explain it, seems mistake in the sentences
- Line 45 please elaborate on the studies for clear meaning
- Line 47, needs to modify grammatically incorrect
- Line 56 IL6ST Please write the full form
- Line 64-65 What was the inference of the findings
- Line 75-76 What do the authors want to say here, please explain
- Figure 5 Oxidative stress both in legend and title of figure 5
- Comments for discussion first para “Better to start with the novel finding of this study”
- In all references, “Journal style” should be modified based on the MDPI journals.
Author Response
Thanks very much for your suggestions. We appreciate them and try to improve our manuscript depend on them. Please see the attachment.
Hope to hear from you as soon as possible.

Author Response
Thanks very much for your suggestions. We appreciate them and try to improve our manuscript depend on them. Please see the attachment.
Here we provide one response letter.
Hope to hear from you as soon as possible.

Round 2
Reviewer 2 Report
α-synuclein Induces Neuroinflammation Injury through the IL6ST-AS/STAT3/HIF-1α Axis
By
Danyu, Han Zhang, Jieli Zhang , Kaixun Huang, Ying Chen, Xiuna Jing, and Enxiang Tao
The manuscript is improved to a good extend after the review, however, there are still some areas where improvement is required.
1. There is text is in red font colour, I assume these are the new lines now inserted in the text. Line 47, 48, and 49: these are the same lines as in line No. 53, 54 and 55.
2. The patient’s details are given in M&M now. Apparently, there is a significant time gap between when the CSF has been aspirated and used for the study. The storage conditions in between are not given. It is critical since the RNA isolation may get seriously affected by storage.
3. When the authors report micro-array experiment all for long-non-coding RNA, why mRNA isolation is described in M&M section?
4. The ethical approval letters are now provided; however, both are in Chinese language, and nothing could be understood from there. Here I rest my argument on this issue, and leave it to the discretion of the Editor in chief to accept this.
5. Lastly, the English still have issues and the draft may need at least one more round of editing especially for the paragraphs that are in red font colour.
